# Household antibiotic use in Malawi: A cross-sectional survey from urban and peri-urban Blantyre

**Eleanor E. MacPherson**[1,2]\*, **John Mankhomwa**[1], **Justin Dixon**[3], **Raymond Pongolani**[1], **Mackwellings Phiri**[1], **Nicholas Feasey**[1,2], **Thomasena O'Byrne**[1,2], **Rachel Tolhurst**[2], **Peter MacPherson**[1,2,3,4]

**1** Malawi-Liverpool-Wellcome Programme, Blantyre, Malawi, **2** Liverpool School of Tropical Medicine, Liverpool, United Kingdom, **3** London School of Hygiene & Tropical Medicine, London, United Kingdom, **4** University of Glasgow, Glasgow, United Kingdom

\* eleanormacp@gmail.com

**Data Availability Statement:** Data and code to reproduce analysis are available at DOI 10.17605/OSF.IO/58EKN.

## Abstract

Antimicrobial resistance (AMR) is a significant threat to public health. Use of antibiotics, particularly in contexts where weaker regulatory frameworks make informal access easier, has been identified as an important driver of AMR. However, knowledge is limited about the ways antibiotics are used in communities in Malawi and sub-Saharan Africa. Between April and July 2021, we undertook a cross-sectional survey of community antibiotic use practices in Blantyre, Malawi. We selected two densely-populated neighbourhoods (Chilomoni and Ndirande) and one peri-urban neighbourhood (Chileka) and undertook detailed interviews to assess current and recent antibiotic use, supported by the innovative "drug bag" methodology. Regression modelling investigated associations with patterns of antibiotic recognition. We interviewed 217 households with a total of 1051 household members. The number of antibiotics recognised was significantly lower among people with poorer formal health care access (people with unknown HIV status vs. HIV-negative, adjusted odds ratio [aOR]: 0.76, 95% CI: 0.77-.099) and amongst men (aOR: 0.83, 95% CI: 0.69–0.99), who are less likely to support healthcare-seeking for family members. Reported antibiotic use was mostly limited to a small number of antibiotics (amoxicillin, erythromycin and cotrimoxazole), with current antibiotic use reported by 67/1051 (6.4%) and recent use (last 6 months) by 440/1051 (41.9%). Our findings support the need for improved access to quality healthcare in urban and peri-urban African settings to promote appropriate antibiotic use and limit the development and spread of AMR.

## Introduction

Drug resistant infections are increasing worldwide and there is global consensus on the need for urgent action to address antimicrobial resistance (AMR) [1]. Optimising the use of antibiotics is one of the central pillars of the World Health Organization (WHO) global action plan to address AMR [2]. The ramifications of AMR are being more acutely felt in low-and-middle-

**Funding:** This study was funded by AMR Cross-Council Initiative through a grant from the Medical Research Council MR/S004793/1. Additionally, this research was funded in whole, or in part, by the Wellcome Trust [Grant numbers: 206545/Z/17/Z and 206575/Z/17/Z]. For the purpose of open access, the author has applied a CC BY public copyright licence to any Author Accepted Manuscript version arising from this submission. The funders had no role in study design, data collection and analysis, decision to publish, or preparation of the manuscript.

**Competing interests:** The authors have declared that no competing interests exist.

income countries (LMICs), where there is a higher burden of infectious diseases and health systems are weaker [3]. In low-income contexts where routine diagnostic microbiology facilities are scarce, data on the epidemiology and burden of AMR is limited [4–6].

In the past two decades, academic and policy research has shown a substantially increased use of antibiotics in LMICs [7,8]. Due to contextual factors including weaker regulatory frameworks and more fragile health systems, antibiotics are often bought over-the-counter, or obtained through informal networks [9,10]. However, research conducted in LMICs has demonstrated that antibiotic use practices, including self-medication, are heterogeneous [11–13]. Do and colleagues (2021) undertook a survey in six countries in Asia and Africa, and found substantial differences in self-medication of antibiotics, with rates lower in Mozambique (8·0%) and South Africa (1·2%) than in Bangladesh (45·7%) and Ghana (36·1%) [14]. Torres and colleagues (2020) did a systematic scoping review of self-medication with antibiotics in 10 low- and middle-income countries, and found a prevalence of self-medication in the preceding year that ranged from 8.1% to 93% across countries [15]. Both these studies found that factors shaping self-medication were complex and included access to, and cost and quality of care at health care facilities [16,17]. The diversity in the prevalence of self-medication of antibiotics in different countries underscores the need for accurate national data to guide antibiotic use and AMR policy and practice.

Global antibiotic consumption assessments have primarily relied on pharmaceutical sales records, but this data is frequently unavailable or unreliable, especially in the African region [18–20]. In LMICs, most academic research has been conducted in secondary or tertiary care facilities. While researchers are increasingly investigating antibiotic use in primary health care and community contexts [21–23], there remain significant gaps in knowledge on antibiotic use practices in community settings, with knowledge particularly limited in countries in Southern and Eastern Africa [24,25]. Given the role reduction in antibiotic use (ABU) is expected to play in reducing AMR, generating robust ABU surveillance data is a necessary first step to inform context-specific interventions [26]. This paper aimed to generate data on current antibiotic use in three residential areas in Blantyre, Malawi.

## Materials and methods

### Ethics statement

Ethical approval was obtained from the College of Medicine Research Ethics Committee, Malawi (approval number: P06/182429) and London School of Hygiene and Tropical Medicine Research Ethics Committee (approval number: 14617). Permission to work in the study communities was granted by the Blantyre District Health Officer. All participants provided written informed consent to participate.

### Study design and population

Data were collected between April and July 2021. We undertook a cross-sectional survey in two densely-populated neighbourhoods (Chilomoni and Ndirande) and one peri-urban neighbourhood (Chileka) in Blantyre, Malawi. Blantyre is a major commercial centre located in the Southern Region of Malawi. The total population of Blantyre District was estimated to be approximately 1 million people in the 2018 Malawi National Census [20], and adult HIV prevalence is estimated to be 18% [21]. Chilomoni and Ndirande are well-established urban neighbourhoods, with poor access to municipal services, and high rates of household poverty. Chileka is located on the periphery of Blantyre and is characterised by a mixture of recently established peri-urban households and households engaged in subsistence farming.

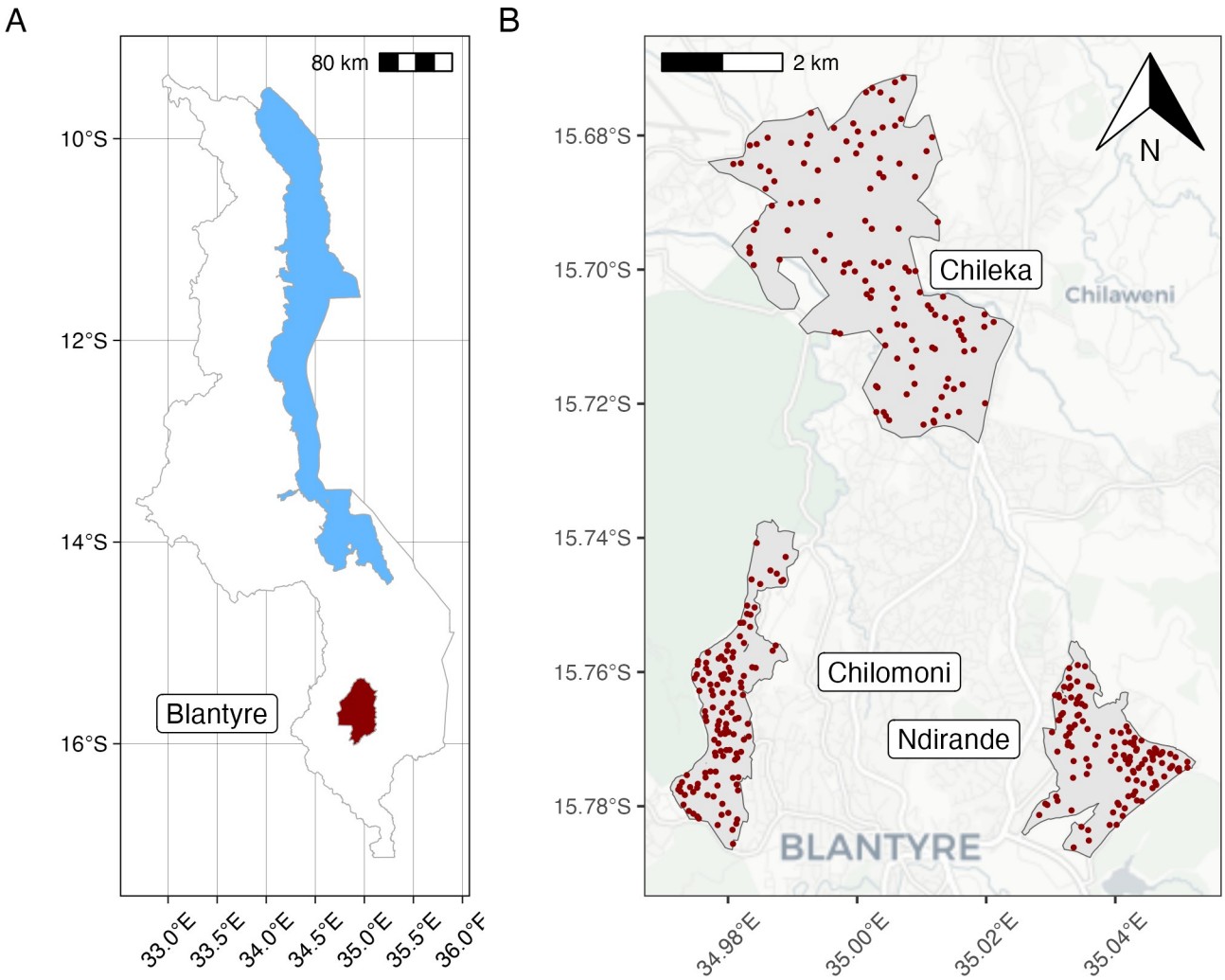

**Fig 1. Location of households sampled for interview.** (A) Malawi boundary data from the United Nations Office for the Coordination of Humanitarian Affairs, Humanitarian Data Exchange (https://data.humdata.org/dataset/geoboundaries-admin-boundaries-for-malawi). (B) Map tile data from OpenStreetMap (https://www.openstreetmap.org/copyright).

Participants eligible to take part in the survey were adults (aged 18 years or older) resident in households in either Chilomoni, Chileka, or Ndirande. We excluded households where there was no adult available to complete questionnaires, or where the household head declined consent to participate.

The unit of sampling was the household. To establish a household sampling frame for Chilomoni and Ndirande, we obtained neighbourhood boundaries and used georeferenced household data from a previous population census of these neighbourhoods conducted as part of a cluster-randomised trial of TB and HIV interventions in 2015 (Fig 1) [22,23]. Households were randomly selected within each neighbourhood, with a replacement list for when households declined participation or were not identified. As Chileka was not part of the 2015 trial census, we selected random sets of GPS coordinates from within the neighbourhood boundaries.

## Procedures

Using GPS coordinates of randomly-selected households in Chilomoni and Ndirande, study fieldworkers navigated to selected households; in Chileka, they navigated to selected locations using GPS coordinates, and identified the nearest household to this point. After identifying the household head and obtaining informed written consent, the fieldworker administered a questionnaire to record demographic and socio-economic characteristics of the household and household head. Data were collected using ODK Collect running on study tablets and uploaded securely in real-time to the Data Server at the London School of Hygiene and Tropical Medicine.

To evaluate household heads' understanding and use of antibiotics, the fieldworker used the "drug bag" method, which comprised of a bag of antibiotics packages and tablets assembled by the team through visiting formal and informal access points for medicines [24]. We adapted this method from anthropological studies we have undertaken exploring medicine use practices in Southern and Eastern Africa [24]. The drug bag was used to facilitate questions to the household head on which antibiotics they recognised (i.e. had ever seen or heard about before), whether antibiotics were currently being used in the household, and more generally about antibiotic use practices in the household (see Supplementary Material). The drug bag helped to overcome linguistic barriers, as 'antibiotic' is a category of medicines that is not readily understood in Malawi. The primary respondent was the household head, however, other members of the family did provide answers during the pile-sorting exercise. The bag included multiple versions of the same antibiotics (active ingredients) to facilitate recognition.

## Statistical methods

We initially set out to survey 75 households in each of two of the study neighbourhoods (Chilomoni and Ndirande), and subsequentially added the third neighbourhood (Chileka) to allow comparison with this peri-urban neighbourhood. Sample size estimates were based on a previous survey, which estimated that ~3% of participants reported using one antibiotic at the time of interview [25]. A sample size of 75 participants per neighbourhood gave a precision of +/- 3% around a point estimate of 3% antibiotic use within each neighbourhood.

We summarised household, and household head characteristics by study neighbourhood using counts and percentages, and means and standard deviations, and compared between groups using Chi-square tests and Fisher's exact tests for categorical data, and Student's T test for continuous data. We used a p-value threshold of $<0.05$ to define a statistically-significant difference between neighbourhoods. Our main study outcomes were antibiotic use and recognition patterns. To investigate this, we summarised the number of household members who reported currently using, or having recently used (within the last 6-months) each antibiotic, and the number of antibiotics recognised by household respondents. To estimate the percentage of the total study population who used and recognised each antibiotic, we divided antibiotics within each category by the total numbers of household members, calculated binomial exact confidence intervals, and compared between study sites. We constructed a multivariable Poisson regression model to investigate associations with the number of antibiotics recognised by household respondents. Analysis was conducted using R v4.1.1 (R Foundation for Statistical Computing, Vienna).

Data and code to reproduce analysis are available at DOI10.17605/OSF.IO/58EKN.

## Financial disclosure statement

This study was funded by AMR Cross-Council Initiative through a grant from the Medical Research Council MR/S004793/1. Additionally, this research was funded in whole, or in part,

by the Wellcome Trust [Grant numbers: 206545/Z/17/Z and 206575/Z/17/Z]. For the purpose of open access, the author has applied a CC BY public copyright licence to any Author Accepted Manuscript version arising from this submission. The funders had no role in study design, data collection and analysis, decision to publish, or preparation of the manuscript.

## Results

In total, 217 households were interviewed and included in this analysis (Table 1). A total of 1051 household members were identified by respondents. Numbers of household residents ranged between 1 and 13, with a median of 5 (interquartile range: 3–6). Numbers of household residents were similar between the three study sites (p = 0.771).

The majority of household respondents were female (80.2%), and the mean respondent age was 38 years. Household and household respondent characteristics were similar in Chilomoni and Ndirande, but there were substantial differences between these neighbourhoods and Chileka. Whereas in Ndirande (76.3%) and Chilomoni (74.6%) most households had electricity supplied to the dwelling, in Chileka only 8.1% had electricity (p<0.001). 78.4% of household respondents in Chileka reported that they were literate, compared to 91.0% in Chilomoni and 82.9% in Ndirande (p = 0.119). Reported coverage of HIV testing was very high and similar across neighbourhoods, with 93.1% of household respondents reporting having ever previously tested for HIV (p = 0.639 for comparison between sites). Self-reported HIV-positive status was 11.1% overall, and was substantially (but not statistically significantly) lower in Chileka (5.4%) compared to Chilomoni (10.4%) and Ndirande (17.1%), p = 0.131. Antiretroviral therapy coverage was 100% among HIV-positive respondents across all three sites.

### Antibiotic recognition

Using the drug bag method, household respondents recognised a median of five antibiotics (IQR: 3–6). The median number of antibiotics recognised in Chileka (4, IQR: 3–6) was slightly lower than in Chilomoni (5, IQR:3–6) or Ndirande (5, IQR: 4–6.25), but this was not statistically significant (p = 0.1176). The most common antibiotics recognised overall by household respondents were: amoxicillin tablets (202/217, 93.1%, 95% CI: 88.9–96.1%); cotrimoxazole tablets (184/217, 84.8%, 95% CI: 79.3–89.3%); and amoxicillin suspension (94/217, 43.3%, 95% CI: 36.6–50.2%). Patterns of antibiotic recognition were broadly similar between the three study sites (Fig 2).

### Associations with recognition of antibiotics

In univariable regression analysis, household respondents from Ndirande (relative risk [RR]: 1.18, 95% CI: 1.02–1.36, compared to Chileka), and from larger households (RR: 1.03, 95% CI: 1.00–1.06 per person increase in household size) recognised a significantly greater number of antibiotics (Table 2). Respondents with unknown HIV status (RR: 0.76, 95% CI: 0.58–0.98) and male household respondents recognised significantly fewer antibiotics than women (0.88, 95% CI: 0.75–1.03) although this was not statistically significant. In multivariable regression, male sex (RR: 0.83, 95% CI: 0.69–0.99) and having unknown HIV status (RR: 0.76, 95% CI: 0.57–0.99) were significantly associated with recognition of fewer antibiotics.

### Antibiotic use

Of the 1051 household members, a total of 67 (6.4%, 95% CI: 5.0–8.0%) were reportedly currently taking one of the antibiotics identified within the drug bag (Table 3). Estimates of household members' current antibiotic use were similar between Chileka (22/355, 6.2%, 95%

**Table 1. Characteristics of households and household respondents, by study site.**

| | Chileka (N = 74) | Chilomoni (N = 67) | Ndirande (N = 76) | Total (N = 217) | P-value |
|---|---|---|---|---|---|
| **Household residents total (median per household, IQR)** | 355 (4, 3–6) | 321 (5, 4–6) | 375 (5, 3–6) | 1051 | 0.771 |
| **Duration residing in site** | | | | | 0.213 |
| Less than one year | 2 (2.7%) | 6 (9.0%) | 7 (9.2%) | 15 (6.9%) | |
| More than one year | 72 (97.3%) | 61 (91.0%) | 69 (90.8%) | 202 (93.1%) | |
| **Household respondent male** | 16 (21.6%) | 17 (25.4%) | 10 (13.2%) | 43 (19.8%) | 0.167 |
| **Household respondent age, Mean (SD)** | 38 (16) | 38 (12) | 37 (13) | 38 (14) | 0.657 |
| **Household respondent occupation** | | | | | < 0.001 |
| Paid domestic worker | 0 (0.0%) | 3 (4.5%) | 4 (5.3%) | 7 (3.2%) | |
| Paid employee | 5 (6.8%) | 4 (6.0%) | 4 (5.3%) | 13 (6.0%) | |
| Self employed | 21 (28.4%) | 27 (40.3%) | 30 (39.5%) | 78 (35.9%) | |
| Student | 1 (1.4%) | 1 (1.5%) | 1 (1.3%) | 3 (1.4%) | |
| Unemployed | 32 (43.2%) | 31 (46.3%) | 37 (48.7%) | 100 (46.1%) | |
| Other | 15 (20.3%) | 1 (1.5%) | 0 (0.0%) | 16 (7.4%) | |
| **Electricity in dwelling** | 6 (8.1%) | 50 (74.6%) | 58 (76.3%) | 114 (52.5%) | < 0.001 |
| **Food sufficient to meet needs** | 38 (51.4%) | 38 (56.7%) | 43 (56.6%) | 119 (54.8%) | 0.759 |
| **Self-rated household poverty†** | | | | | < 0.001 |
| 1 = poorest in neighbourhood | 8 (10.8%) | 5 (7.5%) | 1 (1.3%) | 14 (6.5%) | |
| 2 | 36 (48.6%) | 19 (28.4%) | 24 (31.6%) | 79 (36.4%) | |
| 3 | 27 (36.5%) | 29 (43.3%) | 45 (59.2%) | 101 (46.5%) | |
| 4 | 3 (4.1%) | 14 (20.9%) | 6 (7.9%) | 23 (10.6%) | |
| 5 | 0 (0%) | 0 (0%) | 0 (0%) | 0 (0%) | |
| 6 = richest in neighbourhood | 0 (0%) | 0 (0%) | 0 (0%) | 0 (0%) | |
| **Household respondent highest education level completed** | | | | | 0.044 |
| Missing | 1 | 0 | 0 | 1 | |
| Never been to school | 5 (6.8%) | 3 (4.5%) | 3 (3.9%) | 11 (5.1%) | |
| Primary | 43 (58.9%) | 22 (32.8%) | 29 (38.2%) | 94 (43.5%) | |
| Secondary completed MSCE | 2 (2.7%) | 7 (10.4%) | 8 (10.5%) | 17 (7.9%) | |
| Secondary no MSCE | 22 (30.1%) | 32 (47.8%) | 35 (46.1%) | 89 (41.2%) | |
| Higher | 1 (1.4%) | 3 (4.5%) | 1 (1.3%) | 5 (2.3%) | |
| **Household respondent literate** | 58 (78.4%) | 61 (91.0%) | 63 (82.9%) | 182 (83.9%) | 0.119 |
| **Household respondent self-rated general health** | | | | | 0.538 |
| Missing | 1 | 0 | 2 | 3 | |
| Very poor | 2 (2.7%) | 0 (0.0%) | 0 (0.0%) | 2 (0.9%) | |
| Poor | 7 (9.6%) | 5 (7.5%) | 5 (6.8%) | 17 (7.9%) | |
| Fair | 20 (27.4%) | 19 (28.4%) | 23 (31.1%) | 62 (29.0%) | |
| Good | 31 (42.5%) | 27 (40.3%) | 25 (33.8%) | 83 (38.8%) | |
| Very good | 13 (17.8%) | 16 (23.9%) | 21 (28.4%) | 50 (23.4%) | |
| **Household respondent marital status** | | | | | 0.307 |
| Divorced | 4 (5.4%) | 3 (4.5%) | 6 (7.9%) | 13 (6.0%) | |
| Living together as if married | 0 (0.0%) | 1 (1.5%) | 0 (0.0%) | 1 (0.5%) | |
| Married | 50 (67.6%) | 49 (73.1%) | 59 (77.6%) | 158 (72.8%) | |
| Married but not living together | 2 (2.7%) | 0 (0.0%) | 0 (0.0%) | 2 (0.9%) | |
| Never married | 7 (9.5%) | 3 (4.5%) | 3 (3.9%) | 13 (6.0%) | |
| Polygamous marriage | 0 (0.0%) | 0 (0.0%) | 2 (2.6%) | 2 (0.9%) | |
| Separated | 3 (4.1%) | 2 (3.0%) | 1 (1.3%) | 6 (2.8%) | |
| Widowed | 8 (10.8%) | 9 (13.4%) | 5 (6.6%) | 22 (10.1%) | |

(*Continued*)

**Table 1.** (Continued)

| | Chileka (N = 74) | Chilomoni (N = 67) | Ndirande (N = 76) | Total (N = 217) | P-value |
|---|---|---|---|---|---|
| **Household respondent previously lost spouse to death** | | | | | 0.704 |
| Missing | 1 | 1 | 0 | 2 | |
| Yes | 12 (16.4%) | 12 (18.2%) | 10 (13.2%) | 34 (15.8%) | |
| **Household respondent ever tested for HIV previously** | 68 (91.9%) | 64 (95.5%) | 70 (92.1%) | 202 (93.1%) | 0.639 |
| **Household respondent HIV status** | | | | | 0.131 |
| HIV-negative | 62 (83.8%) | 57 (85.1%) | 58 (76.3% | 177 (81.6%) | |
| HIV-positive | 4 (5.4%) | 7 (10.4%) | 13 (17.1%) | 24 (11.1%) | |
| Unknown | 8 (10.8%) | 3 (4.5%) | 5 (6.6%) | 16 (7.4%) | |
| **Household respondent taking ART (if HIV-positive)** | 4 (100%) | 7 (100%) | 13 (100%) | 24 (100%) | 1.000 |

[†]Self-rated household poverty was ascertained by asking respondents to rate their household's poverty status compared to their neighbours on one of 6 steps, with 1 being the poorest in the neighbourhood, and 6 being the wealthiest in the neighbourhood.

CI: 3.9–9.2%), Chilomoni (20/321, 6.3%, 95% CI: 3.8–9.5%), and Ndirande (25/375, 6.7%, 4.4–9.7%). Only five antibiotics (counted by active ingredient) were identified as being currently taken by household members (Table 2). The antibiotic formulation most commonly reported as currently used was cotrimoxazole tablets (39/1051, 3.7%, 95% CI: 2.7–5.0%), followed by amoxicillin tablets (13/1051, 1.2%, 95% CI: 0.7–2.1%), erythromycin tablets (5/1051, 0.5%, 95% CI: 0.2–1.1%) and amoxicillin suspension (5/1051, 0.5%, 95% CI: 0.2–1.1%).

Household respondents reported that 440 of the 1051 household members (41.9%) had taken an antibiotic in the last 6-months. The antibiotic formulations that had been reportedly most frequently taken in the preceding 6-months were: amoxicillin tablets (126/1051, 12.0%, 95% CI: 10.1%-14.1%); cotrimoxazole tablets (116/1051, 11.0%, 95%CI: 9.2–13.1%); and amoxicillin suspension (38/1051, 3.6%, 95% CI: 2.6–4.9%). Patterns of reported antibiotic use in the preceding 6-months were broadly similar between the three study sites.

## Discussion

The main findings of this detailed cross-sectional survey of community antibiotic use practices in Blantyre Malawi were that current antibiotic use was (67/1051, 6.4%) and recent use (last 6 months) was (440/1051, 41.9%). Both current and recent antibiotic use were limited to a small number of antibiotic formulations, with amoxicillin and cotrimoxazole being the most frequently recognised and used. Both antibiotics are classified as "access" on the WHO's Access, Watch, Reserve (AWaRe) list, meaning they are commonly used to treat infections and should be clinically available at all times [26]. Previous studies in Malawi have shown rapidly emerging bacterial resistance to these antibiotics, threatening clinical and public health utility [27,28]. In contexts such as Malawi, cotrimoxazole is commonly used for chemoprophylaxis for people living with HIV, and with HIV prevalence in the study populations being high, it is not surprising that this was one of the most commonly used and recognised antibiotic. In contrast, the number of antibiotics recognised was significantly lower among people less likely to access formal health services (such as people with unknown HIV status, a strong indicator of delayed healthcare seeking and poor access where testing is widely available), and men, who frequently do not participate in care-seeking activities for families.

Taken together, these findings suggest that, contrary to popular narratives, use of a wide range of antibiotics and antibiotics on the watch or reserve lists was not widespread in these urban and periurban African settings; rather access to a wide range of antibiotics is likely to be

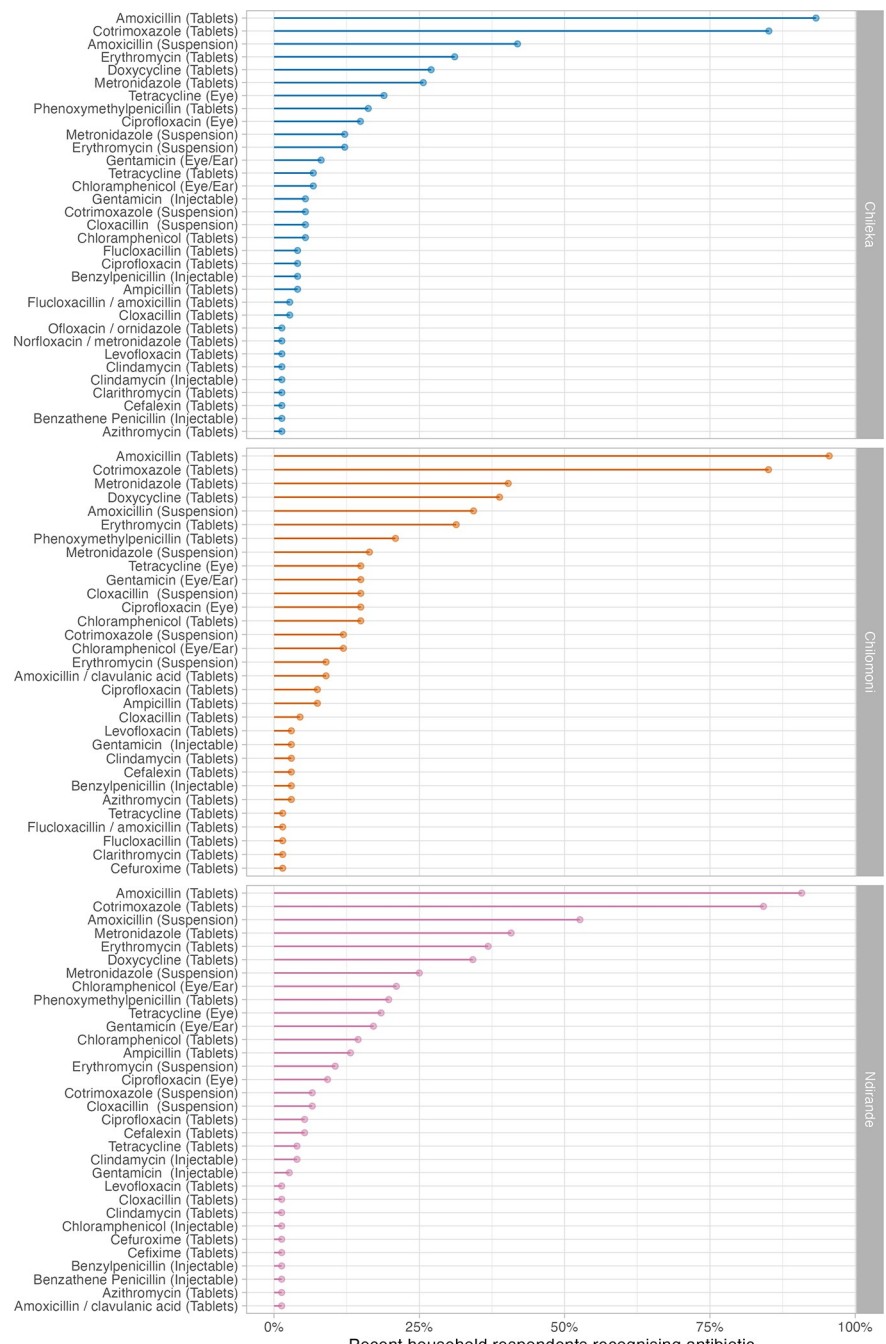

**Fig 2. Household respondent antibiotic recognition, by study site.**

limited for the majority of the population, and potentially not a major contributor to so-called "antibiotic overuse". Nevertheless, antibiotics such as cotrimoxazole, amoxicillin, and erythromycin have broad spectrum action, and dependence on these antibiotics for community management of illness is associated with generation of resistance. Therefore, ABU and AMR guidance must recognise the healthcare and sociological context within which recommendations are made; improvements in universal healthcare access (including treatment of infection)

**Table 2. Associations with number of antibiotics recognised by household respondent.**

| Variable | Univariate relative risk | 95% confidence interval | Multivariable relative risk | 95% confidence interval |
|---|---|---|---|---|
| Site | | | | |
| Chileka | Ref | | Ref | |
| Chilomoni | 1.15 | 0.99–1.34 | 1.10 | 0.94–1.28 |
| Ndirande | 1.18 | 1.02–1.36 | 1.09 | 0.94–1.27 |
| Respondent sex | | | | |
| Female | Ref | | Ref | |
| Male | 0.88 | 0.75–1.03 | 0.83 | 0.69–0.99 |
| Respondent age (per year) | 1.00 | 1.00–1.01 | 1.00 | 1.00–1.01 |
| Household self-rated poverty status | | | | |
| Very poor | Ref | | Ref | |
| Poor | 0.97 | 0.75–1.28 | 0.99 | 0.76–1.31 |
| Somewhat poor | 1.22 | 0.95–1.59 | 1.22 | 0.94–1.61 |
| Somewhat rich | 1.12 | 0.83–1.53 | 1.17 | 0.85–1.62 |
| Respondent HIV status | | | | |
| HIV-negative | Ref | | Ref | |
| HIV-positive | 1.11 | 0.92–1.32 | 1.03 | 0.85–1.25 |
| HIV status unknown | 0.76 | 0.58–0.98 | 0.76 | 0.57–0.99 |
| Number of household members | 1.03 | 1.00–1.06 | 1.02 | 0.99–1.05 |
| Respondent literacy | | | | |
| Respondent illiterate | Ref | | Ref | |
| Respondent literate | 1.10 | 0.93–1.30 | 1.01 | 0.85–1.22 |

should be accompanied by high quality surveillance and antibiotic use data, supported by interrogation of social narratives around the determinants of antibiotic use.

In Malawi and similar settings in Southern and Eastern African countries, primary health care facilities are an important source of community care, with services provided for free [29], but delivered under severe resource constraints with high rates of stockouts of many essential medicines [30,31]. International donor programmes play a significant role in funding services, and infectious diseases have received comparably more resources [32]. The provision of cotrimoxazole by the HIV programme at primary care clinics may explain why it is well recognised and used within households [30]. A multi-site study exploring household antibiotic use in Zimbabwe, Malawi and Uganda identified differences in the profile of ABU, with cotrimoxazole being the most frequently used in rural Malawi, amoxicillin in Harare, Zimbabwe, and metronidazole within informal settlements in Uganda [33]. Most differences in use reflected differences in the configuration of the health system and antibiotic supplies.

In Malawi, gender power relations shape household care seeking practices, with men often presenting at health care facilities critically ill, and having delayed diagnosis of diseases of major public health importance, particularly HIV and tuberculosis [34]. Women spend considerably more time than men seeking care and supporting household members to seek care [35]. These factors are likely to explain why women had better recognition and use of antibiotics. While government health care facilities provide care without user fees, care seeking incurs costs for transportation and loss of time from other livelihood activities [36]. Improving access and removing economic barriers to care seeking, are likely to be important interventions to improve the community management of febrile illness, both to improve recognition of severe illness requiring antibiotics with referral to hospital where required, and to restrict unnecessary ABU.

**Table 3. Antibiotic use by household members.**

| Antibiotic | Chileka (N = 355) n (%, 95%CI) | Chilomoni (N = 321) n (%, 95%CI) | Ndirande (N = 375) n (%, 95%CI) |
|---|---|---|---|
| **Current use** | | | |
| Amoxicillin (Suspension) | 1 (0.3%, 0.0%-1.6%) | 1 (0.3%, 0.0%-1.7%) | 3 (0.8%, 0.2%-2.3%) |
| Amoxicillin (Tablets) | 4 (1.1%, 0.3%-2.9%) | 5 (1.6%, 0.5%-3.6%) | 4 (1.1%, 0.3%-2.7%) |
| Chloramphenicol (Tablets) | | 1 (0.3%, 0.0%-1.7%) | |
| Cotrimoxazole (Tablets) | 13 (3.7%, 2.0%-6.2%) | 10 (3.1%, 1.5%-5.7%) | 16 (4.3%, 2.5%-6.8%) |
| Doxycycline (Tablets) | 1 (0.3%, 0.0%-1.6%) | 1 (0.3%, 0.0%-1.7%) | |
| Erythromycin (Suspension) | | | 2 (0.5%, 0.1%-1.9%) |
| Erythromycin (Tablets) | 3 (0.8%, 0.2%-2.4%) | 2 (0.6%, 0.1%-2.2%) | |
| **Used last 6 months** | | | |
| Amoxicillin (Suspension) | 11 (3.1%, 1.6%-5.5%) | 7 (2.2%, 0.9%-4.4%) | 20 (5.3%, 3.3%-8.1%) |
| Amoxicillin (Tablets) | 44 (12.4%, 9.2%-16.3%) | 39 (12.1%, 8.8%-16.2%) | 43 (11.5%, 8.4%-15.1%) |
| Amoxicillin/clavulanic acid (Tablets) | | 2 (0.6%, 0.1%-2.2%) | |
| Ampicillin (Tablets) | | | 2 (0.5%, 0.1%-1.9%) |
| Benzathine Penicillin (Injectable) | 1 (0.3%, 0.0%-1.6%) | | |
| Benzylpenicillin (Injectable) | | 1 (0.3%, 0.0%-1.7%) | 1 (0.3%, 0.0%-1.5%) |
| Cefalexin (Tablets) | | 1 (0.3%, 0.0%-1.7%) | 1 (0.3%, 0.0%-1.5%) |
| Chloramphenicol (Eye/Ear) | 1 (0.3%, 0.0%-1.6%) | 2 (0.6%, 0.1%-2.2%) | 4 (1.1%, 0.3%-2.7%) |
| Chloramphenicol (Tablets) | | 2 (0.6%, 0.1%-2.2%) | 2 (0.5%, 0.1%-1.9%) |
| Ciprofloxacin (Eye) | 2 (0.6%, 0.1%-2.0%) | 3 (0.9%, 0.2%-2.7%) | 1 (0.3%, 0.0%-1.5%) |
| Ciprofloxacin (Tablets) | 2 (0.6%, 0.1%-2.0%) | 1 (0.3%, 0.0%-1.7%) | 1 (0.3%, 0.0%-1.5%) |
| Clindamycin (Injectable) | | | 1 (0.3%, 0.0%-1.5%) |
| Cloxacillin (Suspension) | 1 (0.3%, 0.0%-1.6%) | | |
| Cloxacillin (Tablets) | 1 (0.3%, 0.0%-1.6%) | | |
| Cotrimoxazole (Suspension) | 1 (0.3%, 0.0%-1.6%) | 1 (0.3%, 0.0%-1.7%) | 1 (0.3%, 0.0%-1.5%) |
| Cotrimoxazole (Tablets) | 47 (13.2%, 9.9%-17.2%) | 30 (9.3%, 6.4%-13.1%) | 39 (10.4%, 7.5%-13.9%) |
| Doxycycline (Tablets) | 6 (1.7%, 0.6%-3.6%) | 7 (2.2%, 0.9%-4.4%) | 12 (3.2%, 1.7%-5.5%) |
| Erythromycin (Suspension) | 3 (0.8%, 0.2%-2.4%) | | 1 (0.3%, 0.0%-1.5%) |
| Erythromycin (Tablets) | 9 (2.5%, 1.2%-4.8%) | 11 (3.4%, 1.7%-6.0%) | 8 (2.1%, 0.9%-4.2%) |
| Gentamicin (Injectable) | | 1 (0.3%, 0.0%-1.7%) | |
| Gentamicin (Eye/Ear) | 1 (0.3%, 0.0%-1.6%) | 3 (0.9%, 0.2%-2.7%) | 4 (1.1%, 0.3%-2.7%) |
| Levofloxacin (Tablets) | | 1 (0.3%, 0.0%-1.7%) | |
| Metronidazole (Suspension) | 4 (1.1%, 0.3%-2.9%) | 3 (0.9%, 0.2%-2.7%) | 7 (1.9%, 0.8%-3.8%) |
| Metronidazole (Tablets) | 9 (2.5%, 1.2%-4.8%) | 9 (2.8%, 1.3%-5.3%) | 13 (3.5%, 1.9%-5.9%) |
| Phenoxymethylpenicillin (Tablets) | 1 (0.3%, 0.0%-1.6%) | 1 (0.3%, 0.0%-1.7%) | 3 (0.8%, 0.2%-2.3%) |
| Tetracycline (Eye) | 1 (0.3%, 0.0%-1.6%) | 2 (0.6%, 0.1%-2.2%) | 4 (1.1%, 0.3%-2.7%) |

Denominators are all reported household members, by site. Binomial exact 95% confidence intervals.

Where cells are blank, household heads reported no use/use in last 6-months of this antibiotic by household members.

Global public engagement campaigns to improve awareness of antimicrobial resistance have focused predominantly on communicating the need to stop people from indiscriminately using antibiotics with insufficient attention paid to the issues of access [37]. Our findings demonstrate frequent use of a very limited number of antibiotic formulations in the preceding six months, however in a context where febrile illness is common and presentation to health care facilities is late, and diagnostics are few, this may represent appropriate prescribing. Indeed, this may instead speak to the need to improve access to and quality of health care services, not limit access, particularly for the most economically disadvantaged groups.

The limitations of this study include the potential for social desirability bias, whereby household respondents may state that they recognised or used antibiotics to meet interviewer's expectations. We mitigated against this by using our innovative and previously validated "drug bag" methodology and experienced field interviewers. The drug bag uses a wide range of antibiotics available in the community; however it is possible that we did not include some important formulations, or that packaging designs have changed, limiting recognition. The concept of an "antibiotic" is not well understood in Malawi and has no specific Chichewa word, potentially hindering accurate recall of use and recognition. We randomly sampled households from a community sampling frame to minimise sampling bias; nevertheless, women were over-represented, perhaps because men may not have been available for interview, or at work. We may therefore have underestimated antibiotic use and recognition. Neighbourhoods were purposively sampled on the basis of availability of sampling frames from previous surveys and demographic/poverty profiles; we did however randomly sample households within neighbourhoods, and this may limit generalisability. We did not collect data on where participants obtained antibiotics from; this should be a priority for future research projects and surveillance.

In conclusion, we found patterns of current and recent antibiotic use and recognition among randomly-sampled household members in Blantyre, Malawi to be limited to a small number of broads-spectrum antibiotic formulations. People known to have poorer access to healthcare reported recognising fewer antibiotics. So-called "antibiotic overuse"–specifically of antibiotics on the watch list–is unlikely to be a major driver of antimicrobial resistance and drug resistance infections in this and similar settings. Rapidly reducing generation of antibiotic resistance and drug resistant infection in low-income settings in Africa will require a shift of focus, away from narratives that blame people in precarious living conditions for "antibiotic overuse" and instead towards holistic approaches that address the underlying systemic drivers of AMR, whilst recognising and supporting antibiotic access within well-functioning health systems.

## Acknowledgments

We are grateful to all study participants who took part in this study and to the DRUM Consortium.

## Author Contributions

**Conceptualization:** Eleanor E. MacPherson, Justin Dixon, Rachel Tolhurst, Peter MacPherson.

**Data curation:** John Mankhomwa, Raymond Pongolani, Mackwellings Phiri, Thomasena O'Byrne.

**Formal analysis:** Eleanor E. MacPherson, Justin Dixon, Peter MacPherson.

**Funding acquisition:** Nicholas Feasey.

**Investigation:** Mackwellings Phiri, Thomasena O'Byrne.

**Methodology:** Eleanor E. MacPherson, Justin Dixon, Raymond Pongolani, Mackwellings Phiri, Rachel Tolhurst, Peter MacPherson.

**Project administration:** Eleanor E. MacPherson, John Mankhomwa, Raymond Pongolani.

**Supervision:** John Mankhomwa, Peter MacPherson.

**Writing – review & editing:** John Mankhomwa, Justin Dixon, Raymond Pongolani, Mackwellings Phiri, Nicholas Feasey, Thomasena O'Byrne, Rachel Tolhurst, Peter MacPherson.

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
