## [Decision Letter · Decision Letter 0]

15 Mar 2023

PGPH-D-23-00080

Household antibiotic use in Malawi: a cross-sectional survey from urban and peri-urban Blantyre

Dear Dr. MacPherson,

Thank you for submitting your manuscript to PLOS Global Public Health. After careful consideration, we feel that it has merit but does not fully meet PLOS Global Public Health’s publication criteria as it currently stands. Therefore, we invite you to submit a revised version of the manuscript that addresses the points raised during the review process.

The review comments can be found at the end of this email, together with any comments from the Editorial Office regarding formatting changes or additional information required to meet the journal's policies at this time.

Please note that your revision may be subject to further review and that this initial decision does not guarantee acceptance.

We look forward to receiving your revised manuscript.

Kind regards,

Mbuzeleni Hlongwa, Ph.D

Academic Editor

Journal Requirements:

1. Please send a completed 'Competing Interests' statement, including any COIs declared by your co-authors. If you have no competing interests to declare, please state "The authors have declared that no competing interests exist". Otherwise please declare all competing interests beginning with the statement "I have read the journal's policy and the authors of this manuscript have the following competing interests:"

2. Please ensure that Funding Information and Financial Disclosure Statement are matched.

3. Please amend your detailed Financial Disclosure statement. This is published with the article. It must therefore be completed in full sentences and contain the exact wording you wish to be published.

4. Please provide separate figure files in .tif or .eps format only and remove any figures embedded in your manuscript file. Please also ensure that all files are under our size limit of 10MB.

5. Fig 1: please (a) provide a direct link to the base layer of the map (i.e., the country or region border shape) and ensure this is also included in the figure legend; and (b) provide a link to the terms of use / license information for the base layer image or shapefile. We cannot publish proprietary or copyrighted maps (e.g. Google Maps, Mapquest) and the terms of use for your map base layer must be compatible with our CC-BY 4.0 license. 

Additional Editor Comments (if provided):

Reviewers' comments:

Reviewer's Responses to Questions

**Comments to the Author**

1. Does this manuscript meet PLOS Global Public Health’s publication criteria? Is the manuscript technically sound, and do the data support the conclusions? The manuscript must describe methodologically and ethically rigorous research with conclusions that are appropriately drawn based on the data presented.

Reviewer #1: Yes

Reviewer #2: Yes

Reviewer #3: Partly

2. Has the statistical analysis been performed appropriately and rigorously?

Reviewer #1: Yes

Reviewer #2: Yes

Reviewer #3: I don't know

3. Have the authors made all data underlying the findings in their manuscript fully available (please refer to the Data Availability Statement at the start of the manuscript PDF file)?

Reviewer #1: Yes

Reviewer #2: Yes

Reviewer #3: Yes

4. Is the manuscript presented in an intelligible fashion and written in standard English?

Reviewer #1: Yes

Reviewer #2: Yes

Reviewer #3: No

5. Review Comments to the Author

Reviewer #1: Reviewer Comments

Thank you very much for the opportunity to review this manuscript. Unregulated antibiotics use by the community in low- and middle-income countries is a major factor in the development of AMR. The cross-sectional survey reported in this manuscript adds to the body of knowledge in this subject area and I would like to thank the authors for this work.

There are however few comments and questions I would like the authors to address:

Methods

1. What informed the selection of the study sites? How did you deal with possible selection bias?

2. Why were two neighbouring urban sites (Chilomoni and Ndirande) comapared with one peri-urban site (Chileka)? For a better comparison of community antibiotic use in Malawi, the inclusion of a typical rural site could have been done (i.e., one each of urban, peri-urban and rural randomly selected).

Results

3. Table 1: Details of the Self-rated household poverty scales 2 – 5 are missing.

4. Under Antibiotic use: Lines 199 – 201 – The sentence “The bag 200 included multiple versions of the same antibiotics (active ingredients) to facilitate 201 recognition” is not a finding. It should have been included in the explanation of the “drug bag” in the Methods section.

Discussion and Conclusion – No comments

Reviewer #2: Thanks for the opportunity to review this well-written and timely manuscript that addresses an important question in the development of AMR in low-and middle-income countries, and the impact of community-level antibiotic use. The manuscript has many strengths including its thorough sampling methodology, novel use of the drug-bag methodology, and statistical analysis. The manuscript highlights the important issue of both overuse and limited access to antimicrobials in these resource-poor contexts. Data of these kinds are severely lacking from contexts such as Malawi and therefore I would like to recommend this manuscript for publication with a few points that the authors may want to consider.

1. Did you collect data on where households were acquiring their antimicrobial? Studies from south-Asia demonstrate extensive use of private pharmacies without the need for formal healthcare consultation, which is important when it comes to regulation. Patterns of healthcare utilisation are quite different in southern/eastern Africa, and I think some comment on this in the discussion would be instructive.

2. From published surveillance studies performed in Blantyre (e.g., Musicha et al, Lancet ID and Meiring et al, Lancet GH, there are others) it is striking how the commonest causes of bacterial infection are resistant to the antibiotics that people are using in their homes that you have highlighted through this study. The recognition of and use of ciprofloxacin is surprisingly low, and then amoxicillin with clavulanate, macrolides and 3rd generation cephalosporins are completely absent. I think it will strengthen your argument on lack of access to appropriate healthcare provision if you connect the dots between your findings and the results of these surveillance studies.

Reviewer #3: 6 Include Malawi because the study is targeting Malawi.

31 WHO is appearing for the first time so write in full and put that in brackets

41 Where heterogeneous in this case shall mean? And there are examples coming up I guess to illustrate the heterogeneity, but there is no linkage to show that. 'For example' can help sorting out that issue.

47 'Both these studies' should be checked.

41 This line and onwards shows that the study is narrowing down to self-medication. However the approach to that has no justification why the study focused on that area. It would be better to state a few of such bad practices and single out the 'self-medication' and state why.

53 The introduction goes back to general issues about antimicrobial use practices. So it's difficult to tell from the introduction about its focus. So either talk in general terms of the uses or narrow down to self-medication.

61 'This paper aimed to generate'....check this sentence.

62 Is there any reason for choosing the 3 sites? Of all the areas in Malawi, Blantyre or urban areas? Of course line 78 tries to explain this, but I do not think the explanation makes it any different from other place or act as justification because the stated reasons are not unique to the areas mentioned, unless if there is data to back that up or if there is, the sentence to should rewritten to bring it out.

78 In other parts, citations are put after a full stop and here it's before. So there must be consistency. One one way and replicate throughout the document.

90 Which census is that? 2018 or 2008 census? How did you get the geo-referenced data?

77 There is a mention of HIV prevalence. Is there any relationship of the study to it or it is just mentioned for the reader's attention? This is not clear although I have seen it coming out in the discussion section. If the HIV was the basis of the study, or selection of the study sites, it should have been stated.

101 How did you select the households? Did you select all those geo-referenced in the previous studies or selected some out of those? I expect that your sample sizes were different and yours might have been smaller than those earlier studies unless stated otherwise. How many people belonged to all the households and how many were not eligible due to either age or declined consent? Since you have mentioned that you involved all members of a household but few were left out.

109-118 You have mentioned how you recruited the head of household, but not the other members in the households. How were the other members recruited? And how did you ensure that there is no response contamination from previous respondent of the household on the new respondents of the same family? And how did you ensure privacy of the respondents from other family members?

118 How did you deal with the issue of repackaged antibiotics since some of them are not sold in their original packs? Are there no colour mix-ups between antibiotics of different APIs in Malawi that you would be sure that colour would be enough for someone to remember previous use of that medicine?

In the comparisons of the neighborhood, how did you deal with the issue of different education backgrounds of the respondent's as well as levels of income? Because these factors can affect the comparative analysis. Or how did you deal with these confounders?

175 Talking about household respondents, yet the population in the respondent's is 217. So were all the responses in a household counted as 1 or each was independent? Because if you talk about household respondents, I expect a population bigger than 217, because that's what it is in the text, 1051. Or are you giving the responses of household heads only this time?

Figure 2. Were there no capsules in the drug bag? Or the word tablets is being used interchangeably with capsules because mostly antibiotics are packed in capsules.

Antibiotics brands differ from place to place and period to period. How did you deal with that issue? Because the time of putting up a drug bag might differ from the time the respondents used a particular drug which might also affect its presence during the questioning. Was there a time limit to which the respondents recruited had used the drugs because the longer the time, the higher the chances of forgetting drug colour or product name especially when you don't anticipate and future use of such information.

218 Were there any attempts to relate the suspension use to number of children in the households?

230 There is an attempt to link cotrimoxazole to HIV, yet this drug is used for many diseases as well or is this the only indication for this drug in Malawi Treatment Guideline? And is it used strictly for that purpose in Malawi and that it can only be accessed in strict ways? How was this arrived at to be attributed to HIV? Were there any questions related to the reasons for using a particular antibiotic? Or were there any statistical tests that showed this possibility? If so, highlight them in the discussion by putting a table reference. Amoxicillin was also found to be used widely. Why is it also not attributed to the HIV? Dont Amoxicillin and Cotrimoxazole have similar uses in one way or the other? Take note that antibiotics are also widely known to be used in colds and flu apart from their intended uses.

232-236 There is an extrapolation of the results that is not supported by the results or any literature at least in this manuscript. Failure to recognise a drug cannot translate to delayed health seeking behaviour. People use antibiotics only when they are sick or for prophylaxis. So if people with unknown HIV status are not using them, will that mean then that they might be HIV positive and therefore delayed health seeking behaviour?

Did you make attempts to look at access issues? Like how they got the antibiotics? Were the antibiotics used after getting prescription or just buying on their own? What about the source? Distances to the hospital? Or access to drug stores or pharmacies? I think these would have helped in generating more answers to the questions of your study. Otherwise, use alone cannot be an issue or it can be an issue if we look at reasons for use and how the drugs are accessed.

252 add 'for' after provided.

251-261 Is talking about access of cotrimoxazole through primary health care system, which has not been mentioned in this studys data collection or analyses (If it is mentioned and I have missed it, please include a table citation in the discussion). Where is cotrimoxazole use being attributed to healthcare facilities in the results? That is why I mentioned earlier that how they accessed the antibiotics and reasons thereof would have been relevant.

263 Another reason why women were better at recognising the medicines would be their frequent encounter with the medicines as they mostly also take care of children particularly under Five children who are mostly prescribed with antibiotics for almost every disease due to lack of laboratory facilities in most health centres. Flus and colds are also mostly prescribed antibiotics and flus and colds are very common in children.

277-283 This statement is built on no results from this study or no cited literature that this is the case in the areas of the study. This supports my earlier suggestions that reasons for using the antibiotics stated should have been recorded.

6. PLOS authors have the option to publish the peer review history of their article (what does this mean?). If published, this will include your full peer review and any attached files.

**Do you want your identity to be public for this peer review?** For information about this choice, including consent withdrawal, please see our Privacy Policy.

Reviewer #1: No

Reviewer #2: **Yes: **James E. Meiring

Reviewer #3: No

---

## [Decision Letter · Decision Letter 1]

6 Jul 2023

Household antibiotic use in Malawi: a cross-sectional survey from urban and peri-urban Blantyre

PGPH-D-23-00080R1

Dear Dr MacPherson,

We are pleased to inform you that your manuscript 'Household antibiotic use in Malawi: a cross-sectional survey from urban and peri-urban Blantyre' has been provisionally accepted for publication in PLOS Global Public Health.

Best regards,

Mbuzeleni Hlongwa, Ph.D

Academic Editor

Reviewer Comments (if any, and for reference):

Reviewer's Responses to Questions

**Comments to the Author**

1. If the authors have adequately addressed your comments raised in a previous round of review and you feel that this manuscript is now acceptable for publication, you may indicate that here to bypass the “Comments to the Author” section, enter your conflict of interest statement in the “Confidential to Editor” section, and submit your "Accept" recommendation.

Reviewer #2: All comments have been addressed

Reviewer #3: All comments have been addressed

2. Does this manuscript meet PLOS Global Public Health’s publication criteria? Is the manuscript technically sound, and do the data support the conclusions? The manuscript must describe methodologically and ethically rigorous research with conclusions that are appropriately drawn based on the data presented.

Reviewer #2: Yes

Reviewer #3: Yes

3. Has the statistical analysis been performed appropriately and rigorously?

Reviewer #2: Yes

Reviewer #3: I don't know

4. Have the authors made all data underlying the findings in their manuscript fully available (please refer to the Data Availability Statement at the start of the manuscript PDF file)?

Reviewer #2: Yes

Reviewer #3: Yes

5. Is the manuscript presented in an intelligible fashion and written in standard English?

Reviewer #2: Yes

Reviewer #3: Yes

6. Review Comments to the Author

Reviewer #2: I am happy with the responses of the authors and can recommend this article for publication.

Reviewer #3: The authors have addressed the comments adequately.

7. PLOS authors have the option to publish the peer review history of their article (what does this mean?). If published, this will include your full peer review and any attached files.

**Do you want your identity to be public for this peer review?** For information about this choice, including consent withdrawal, please see our Privacy Policy.

Reviewer #2: **Yes: **James Meiring

Reviewer #3: No

<quillbot-extension-portal></quillbot-extension-portal>